# Event-Triggered Kalman Filter and Its Performance Analysis

**DOI:** 10.3390/s23042202

**Published:** 2023-02-15

**Authors:** Xiaona Li, Gang Hao

**Affiliations:** 1School of Electronic Engineering, Heilongjiang University, Harbin 150080, China; 2Key Laboratory of Information Fusion Estimation and Detection, Harbin 150080, China

**Keywords:** event trigger, threshold setting, accuracy comparison, state estimation

## Abstract

In estimation of linear systems, an efficient event-triggered Kalman filter algorithm is proposed. Based on the hypothesis test of Gaussian distribution, the significance of the event-triggered threshold is given. Based on the threshold, the actual trigger frequency of the estimated system can be accurately set. Combining the threshold and the proposed event-triggered mechanism, an event-triggered Kalman filter is proposed and the approximate estimation accuracy can also be calculated. Whether it is a steady system or a time-varying system, the proposed algorithm can reasonably set the threshold according to the required accuracy in advance. The proposed event-triggered estimator not only effectively reduces the communication cost, but also has high accuracy. Finally, simulation examples verify the correctness and effectiveness of the proposed algorithm.

## 1. Introduction

The growing demands of communications, navigation positioning, fault detection, and environmental monitoring have spawned wireless sensor network systems [1,2,3,4,5]. In wireless sensor networks, due to the limitation of bandwidth and energy, frequent data transmission is unfavorable, which not only increases the communication costs, but also induces some negative network phenomena, such as packet loss, network delay, etc. In order to reduce the waste of resources in the operation of the equipment and ensure the quality of communication, a better transmission mechanism is worth seeking. Unlike traditional time-triggered mechanisms, an event-triggered one transmits information only when needed. It effectively alleviates the problems of network congestion and waste of resources, and has attracted widespread attention from scholars [6,7,8,9,10].

As early as 1983, Ho et al. first introduced the concept of event trigger into discrete systems [11]. Later, in the development of event-triggered filter estimation, a trigger mechanism named send-on-delta (SOD) emerges. The trigger mechanism is based on differences in the measured signals, and when the signal deviates from the delta, the communication is triggered. It is one of the classic solutions to solve redundant data transmission in wireless sensor networks [12]. Subsequently, in [13], using estimation variance as a trigger condition, Trimpe et al. correlated trigger decision with estimator performance, and estimation algorithms for distributed sensor nodes that can be applied in two cases were designed, and another popular trigger mechanism was proposed. Based on the innovation and its covariance information, Wang N.et al. proposed an event-triggered sequence fusion estimator. It is an improvement of the SOD mechanism, which solves the problem of event-triggered estimators of observation noise under the linear minimum variance criterion [14]. Zhang et al. introduced the event trigger mechanism into the H∞ filtering framework. The trigger conditions depend on the current measurement and the difference between the latest transmission, which significantly saves communication resources, and the filtering error has proved to be asymptotically stable [15]. In [16], the event-triggered mechanism under malicious denial-of-service attacks is studied, and estimating performance and network communication are analyzed.

The trigger mechanisms used in the above works are all static; that is, a constant is used as the trigger threshold [17]. With the in-depth study of event triggers, the setting of the threshold is no longer conservative and dynamic event trigger mechanisms appeared [18]. In [19], taking into account the unknown input of the network control system, an adaptive threshold including system state vector and neural network weight estimation is designed, and the stability of the system is rigorously proved by using the Lyapunov stability theory. In [20], for the multi-agent system, event-triggered mechanisms in the centralized and distributed cases are developed, and the dynamic threshold of the distributed system is set by introducing internal variables and exponential functions, which greatly reduces the transmission burden. For uncertain systems with transmission delays, a distributed transmission scheme based on a dynamic event trigger is proposed in [21]. Different from [20], the trigger time series of the system is determined by the dynamic event variable rules. In [22], a novel decentralized trigger condition is designed, whose trigger threshold is constructed by the rate of change, and the Zeno behavior in the trigger process is avoided by adding a corresponding constant term.

In fact, the selection of the event-triggered threshold depends on the preset accuracy and trigger rate, but there are few works on the accuracy range and trigger rate at a certain threshold. Compared with the traditional trigger mechanism, the one proposed in this paper can give a quantitative relationship between trigger threshold, communication rate, and estimation error. Clearly, it is important for wireless sensors to clarify the relationship between the communication frequency and the corresponding estimation accuracy range. At the same time, the trigger mechanism with better performance at the same trigger rate is sought. For these reasons, the motivation for our study came into being. The main contributions of this paper are summarized as follows:For wireless sensors, an event-triggered mechanism is proposed. The proposed event-triggered mechanism is based on a normal distribution constructed from the predicted and filtered differences. When the constructed normal distribution trigger function exceeds a tolerable threshold, the filtering results are transmitted.The theoretical trigger probability and estimation accuracy under the proposed trigger mechanism are derived. For linear time-invariant systems, the Riccati equation can be used to obtain the estimation error variance of the event-triggered estimator. For linear time-varying systems without steady-state estimation, an approximate estimation accuracy can be obtained. Therefore, their trigger threshold can be set according to the precision requirements.The simulation verifies the correctness of the proposed theorems and inferences. Compared with several types of event trigger mechanisms in the existing literature [14,22], the trigger mechanism proposed in this paper has higher estimation accuracy and better performance under the same trigger rate.

Notation: $R^{n}$ denotes the n-dimensional Euclidean space, $R^{n\times m}$ denotes the set of n by m real-valued matrices, ‘E’ denotes the mathematical expectation, $M^{T}$ and $M^{-1}$ are the transpose and inverse of matrix M, respectively, chol(P) is the Cholesky decomposition of matrix P, tr(M) is the trace of matrix M, $I_{n}$ is the n-dimensional identity matrix, $N(\mu ,\delta^{2})$ denotes the Gaussian distribution with mean μ and variance $\delta^{2}$, $\delta_{kt}$ is the Kronecker delta function $\left(\delta_{\mathrm{tt}}=1,\delta_{tk}=0\;(t\neq k)\right)$, ${\tilde{x}}^{}{}_{k|k-\tau }=x_{k}-{\hat{x}}^{}{}_{k|k-\tau }$ is the state estimation error, and ${\tilde{z}}^{}{}_{k|k-\tau }=z_{k}-{\hat{z}}^{}{}_{k|k-\tau }(\tau =0,1,\cdots ,k-1)$ is the measurement prediction error.

## 2. Problem Formulation

Consider the following linear system:(1)$$x_{k+1}=\Phi x_{k}+\Gamma w_{k}$$
(2)$$z_{k}=Hx_{k}+v_{k}$$
where k is discrete time, $x_{k+1}\in R^{n}$ is the state vector, $z_{k}\in R^{m}$ is the measurement vector, $\Phi \in R^{n\times n}$ is the state transition matrix, $\Gamma \in R^{n\times m}$ is the noise transfer matrix, $H\in R^{m\times n}$ is the measurement matrix, $w_{k}$ is the process noise, and $v_{k}$ is the measurement noise.

**Assumption** **1.**
*$w_{k}$ and $v_{k}$ are uncorrelated white Gaussian noise processes with zero means and variances $Q_{k}$ and $R_{k}$, respectively,*

(3)
$$E\left[\left(\begin{array}{l} w_{k}\\ v_{k} \end{array}\right)\left(w_{t}^{T}\;v_{t}^{T}\right)\right]=\left[\begin{array}{ll} Q_{k}\; & 0\\ 0 & R_{k} \end{array}\right]\delta_{kt}$$



**Assumption** **2.**
*The initial state $x_{0}$ is dependent on $w_{k}$ and $v_{k}$, and satisfies:*

(4)
$$E[x_{0}]=\mu_{0},\;E[(x_{0}-\mu_{0}){\left(x_{0}-\mu_{0}\right)}^{T}]=P_{0}$$



### 2.1. Threshold Selection of Event-Triggered Mechanism

Taking into account the characteristics of intelligence and the low power consumption of current sensors, as well as the advantages of event-triggered mechanisms based on state estimation, an event-triggered Kalman filter based on the state estimation is designed. The smart sensor node is responsible for continuous filtering and judgment. If the trigger condition is not satisfied, the prediction actions will be performed by the center; otherwise, the filter estimation will be sent by the smart sensor node and the target states will be updated by the receiving center. This structure can effectively reduce communication redundancy.

**Theorem** **1.***For the system in Equations (1) and (2), the test statistics obey the following distribution:*(5)$$\gamma_{k}={\left(\mathrm{chol}(P_{k,\tau }^{x})\right)}_{}^{-1}({\hat{x}}_{k|k-\tau }-{\hat{x}}_{k|k})\sim N\left(0,I_{n}^{}\right)\;$$*where*(6)$${\hat{x}}_{k|k}={\hat{x}}_{k|k-1}+K_{k}{\tilde{z}}_{k|k-1}$$(7)$$K_{k}=P_{k|k-1}H^{T}{\left[HP_{k|k-1}H^{T}+R_{k}\right]}^{-1}$$(8)$${\tilde{z}}_{k|k-1}=z_{k}-H{\hat{x}}_{k|k-1}$$(9)$${\hat{x}}_{k|k-1}=\Phi {\hat{x}}_{k-1|k-1}$$(10)$$P_{k|k-1}^{}=\Phi P_{k-1|k-1}^{}\Phi^{T}+\Gamma Q_{k-1}^{}{\left(\Gamma \right)}^{T}$$${\hat{x}}_{k|k-\tau }$ *is the*
 τ−step *Kalman prediction:*
(11)$${\hat{x}}_{k|k-\tau }=\Phi^{\tau -1}{\hat{x}}_{k-\tau +1|k-\tau },\tau \geq 1\;$$
*and the second moment* $P_{k,\tau }^{x}$ *of*  $\left({\hat{x}}_{k|k-\tau }-{\hat{x}}_{k|k}\right)$ *can be written as:*
(12)$$\begin{array}{l} P_{k,\tau }^{x}=E\{({\hat{x}}_{k|k-\tau }-{\hat{x}}_{k|k}){\left({\hat{x}}_{k|k-\tau }-{\hat{x}}_{k|k}\right)}_{}^{T}\}\\ \;\;\;\;\;\;\;=P_{k|k}-P_{k|k-\tau }-P_{k|k-\tau }^{\left(1\right)}-P_{k|k-\tau }^{\left(2\right)}-\cdots -P_{k|k-\tau }^{\left(\tau \right)}-{\left(P_{k|k-\tau }^{\left(1\right)}\right)}_{}^{T}-{\left(P_{k|k-\tau }^{\left(2\right)}\right)}_{}^{T}-\cdots -{\left(P_{k|k-\tau }^{\left(\tau \right)}\right)}_{}^{T} \end{array}$$
*where the filtering error variance*
 $P_{k|k}$
*is:*
(13)$$P_{k|k}=[I_{n}-K_{k}H]P_{k|k-1}\;$$
*the prediction error variance*
 $P_{k|k-\tau }$ *in Equation (12) is calculated by:*
(14)$$P_{k|k-\tau }=\Phi^{\tau -1}P_{k-\tau +1|k-\tau }{\left(\Phi^{\left(\tau -1\right)}\right)}_{}^{T}+\sum_{j=2}^{\tau }\Phi^{\tau -j}\Gamma Q_{k-\tau +j-1}^{}\Gamma_{}^{T}\Phi^{\left(\tau -j\right)}{}^{T},\tau \geq 2\;$$
 $P_{k|k-\tau }^{\left(m\right)}(m=1,\cdots ,\tau )$ *in Equation (12) is calculated by:*
(15)$$P_{k|k-\tau }^{\left(m\right)}=-\Phi^{\tau -1}P_{k-\tau +1|k-\tau }{\left(A_{m}\right)}^{T}H^{T}K_{k-\tau +m}^{T}{\left(\Phi^{\tau -m}\right)}^{T}+B\;$$
*and the solution steps for*
 $P_{k|k-\tau }^{\left(m\right)}(m=1,\cdots ,\tau )$ *in Algorithm 1 is:*
**Algorithm 1: *Solution steps for*** $P_{k|k-\tau }^{\left(m\right)}$***Initialize:***  $A_{1}=I_{n},C_{0}=I_{n}.$***Iterate:******for***  m=1:τ$A_{m+1}=\Phi A_{m}-\Phi K_{k-\tau +m}^{}HA_{m}$***end******if*** τ=1,B=0   ***else*** $\tau =2,m=2,B=-\Phi^{\tau -m}\Gamma Q_{k-\tau +m+1}\Gamma^{T}I_{n}^{T}H^{T}K_{k-\tau +m}^{T}{\left(\Phi^{\tau -m}\right)}^{T}$  ***else*** $B_{m}^{}=0$***for*** m=3:τ  h=m−2, p=m−2 ***for*** ξ=1:m−2 ***for*** ∂=0:h−1 $C_{\partial +1}=\Phi C_{\partial }-\Phi K_{k-\tau +m-h+\partial }^{}H^{}C_{\partial }$ ***end*** $B_{m}^{\psi }=-\Phi^{\tau -m+p}\Gamma Q_{k-\tau +m-p-1}\Gamma^{T}C_{h}^{T}H^{T}K_{k-\tau +m}^{T}{\left(\Phi^{\tau -m}\right)}^{T}$ h=h−1, p=p−1 ***end***$B_{m}^{}=B_{m}^{}+B_{m}^{\psi }$ ***end***$B=B_{m}-\Phi^{\tau -m}\Gamma Q_{k-\tau +m+1}\Gamma^{T}I_{n}^{T}H^{T}K_{k-\tau +m}^{T}{\left(\Phi^{\tau -m}\right)}^{T}$ ***end***                     $P_{k|k-\tau }^{\left(m\right)}=-\Phi^{\tau -1}P_{k-\tau +1|k-\tau }{\left(A_{m}\right)}^{T}H^{T}K_{k-\tau +m}^{T}{\left(\Phi^{\tau -m}\right)}^{T}+B$


**Proof.** Due to ${\hat{x}}_{k|k-\tau }\in L(z_{1},\cdots ,z_{k-\tau })$ (L is the linear manifold of $z_{1},\cdots ,z_{k-\tau }$) and ${\hat{x}}_{k|k}\in L($$z_{1},\cdots ,z_{k})$ in Equation (5), $\gamma_{k}\in L(z_{1},\cdots ,z_{k})$ obeys the Gaussian distribution.
(1)The proof of the mean value of $\gamma_{k}^{}$. Since ${\hat{x}}_{k|k-\tau }$ and ${\hat{x}}_{k|k}$ are unbiased,
(16)$$E\{\gamma_{k}^{}\}=E\{{\left(\mathrm{chol}(P_{k,\tau }^{x})\right)}_{}^{-1}({\hat{x}}_{k|k-\tau }-{\hat{x}}_{k|k})\}=E\{{\left(\mathrm{chol}(P_{k,\tau }^{x})\right)}_{}^{-1}\}(E\{{\hat{x}}_{k|k-\tau }\}-E\{{\hat{x}}_{k|k}\})=0$$(2)The proof of the $P_{k,\tau }^{x}$ in Equation (12). Based on the Kalman filter:
(17)$$\begin{array}{l} {\hat{x}}_{k|k}={\hat{x}}_{k|k-1}+K_{k}{\tilde{z}}_{k|k-1}=\Phi {\hat{x}}_{k-1|k-1}+K_{k}{\tilde{z}}_{k|k-1}=\Phi^{2}{\hat{x}}_{k-2|k-2}+\Phi K_{k-1}{\tilde{z}}_{k-1|k-2}+K_{k}{\tilde{z}}_{k|k-1}\\ =\Phi^{\tau }\;{\hat{x}}_{k-\tau |k-\tau }+\Phi^{\tau -1}K_{k-\tau +1}{\tilde{z}}_{k-\tau +1|k-\tau }+\Phi^{\tau -2}K_{k-\tau +2}{\tilde{z}}_{k-\tau +2|k-\tau +1}+\ldots +K_{k}{\tilde{z}}_{k|k-1}\\ ={\hat{x}}_{k|k-\tau }+\sum_{i=1}^{\tau }\Phi^{\tau -i}K_{k-\tau +i}{\tilde{z}}_{k-\tau +i|k-\tau +i-1} \end{array}$$
estimated error variance matrix $P_{k,\tau }^{x}$ is computed by:(18)$$\begin{array}{l} P_{k,\tau }^{x}=E\{({\hat{x}}_{k|k-\tau }-{\hat{x}}_{k|k}){\left({\hat{x}}_{k|k-\tau }-{\hat{x}}_{k|k}\right)}^{T}\}\\ =E\{[({\hat{x}}_{k|k-\tau }-x_{k})+(x_{k}-{\hat{x}}_{k|k})]{\left[({\hat{x}}_{k|k-\tau }-x_{k})+(x_{k}-{\hat{x}}_{k|k})\right]}^{T}\}\\ =P_{k|k-\tau }+P_{k|k}-E\left\{({\hat{x}}_{k|k-\tau }-x_{k}){\left({\hat{x}}_{k|k}-x_{k}\right)}^{T}\right\}-E\left\{({\hat{x}}_{k|k}-x_{k}){\left({\hat{x}}_{k|k-\tau }-x_{k}\right)}^{T}\right\} \end{array}$$
From Equations (17) and (18), $E\left\{({\hat{x}}_{k|k-\tau }-x_{k}){\left({\hat{x}}_{k|k}-x_{k}\right)}^{T}\right\}$ can be written as:(19)$$\begin{array}{l} E\left\{({\hat{x}}_{k|k-\tau }-x_{k}){\left({\hat{x}}_{k|k}-x_{k}\right)}^{T}\right\}\\ =E\left\{({\hat{x}}_{k|k-\tau }-x_{k}){\left({\hat{x}}_{k|k-\tau }-x_{k}+\sum_{i=1}^{\tau }\Phi^{\tau -i}K_{k-\tau +i}{\tilde{z}}_{k-\tau +i|k-\tau +i-1}\right)}^{T}\right\}\\ =P_{k|k-\tau }+E\left\{\sum_{i=1}^{\tau }({\hat{x}}_{k|k-\tau }-x_{k})[\Phi^{\tau -i}K_{k-\tau +i}^{}(H{\tilde{x}}_{k-\tau +i|k-\tau +i-1}+v_{k-\tau +i}){]}_{}^{T}\right\}\\ +E\left\{({\hat{x}}_{k|k-\tau }-x_{k}){\left[\Phi^{\tau -2}K_{k-\tau +2}(H{\tilde{x}}_{k-\tau +2|k-\tau +1}+v_{k-\tau +2})\right]}^{T}\right\}\\ +\ldots +E\left\{({\hat{x}}_{k|k-\tau }-x_{k}){\left[K_{k}(H{\tilde{x}}_{k|k-1}+v_{k})\right]}^{T}\right\}=P_{k|k-\tau }+P_{k|k-\tau }^{\left(1\right)}+P_{k|k-\tau }^{\left(2\right)}+\cdots +P_{k|k-\tau }^{\left(\tau \right)} \end{array}$$
since $w_{k-i}$ is white noise and independent of $v_{k-\tau +i}$, $P_{k|k-\tau }^{\left(1\right)}$, $P_{k|k-\tau }^{\left(2\right)}$, $P_{k|k-\tau }^{\left(3\right)}$ in Equation (19) are respectively calculated as:(20)$$\begin{array}{l} P_{k|k-\tau }^{\left(1\right)}=E\left\{({\hat{x}}_{k|k-\tau }-x_{k}){\left[\Phi^{\tau -1}K_{k-\tau +1}(H{\tilde{x}}_{k-\tau +1|k-\tau }+v_{k-\tau +1})\right]}^{T}\right\}\\ =E\{\left[\Phi^{\tau -1}({\hat{x}}_{k-\tau +1|k-\tau }-x_{k-\tau +1})-L(w_{k-\tau +1}\ldots w_{k-1})\right]{\left[\Phi^{\tau -1}K_{k-\tau +1}(H{\tilde{x}}_{k-\tau +1|k-\tau }+v_{k-\tau +1})\right]}^{T}\}\\ =-\Phi^{\tau -1}P_{k-\tau +1|k-\tau }H^{T}K_{k-\tau +1}^{T}{\left(\Phi^{\tau -1}\right)}^{T} \end{array}$$
(21)$$\begin{array}{l} P_{k|k-\tau }^{\left(2\right)}=E\left\{({\hat{x}}_{k|k-\tau }-x_{k}){\left[\Phi^{\tau -2}K_{k-\tau +2}(H{\tilde{x}}_{k-\tau +2|k-\tau +1}+v_{k-\tau +2})\right]}^{T}\right\}\\ =E\{\left[\Phi^{\tau -1}({\hat{x}}_{k-\tau +1|k-\tau }-x_{k-\tau +1})-L(w_{k-\tau +1}\cdots w_{k-1})\right]{\left[\Phi^{\tau -2}K_{k-\tau +2}(H{\tilde{x}}_{k-\tau +2|k-\tau +1}+v_{k-\tau +2})\right]}^{T}\}\\ =-\Phi^{\tau -1}P_{k-\tau +1|k-\tau }{\left(I_{n}^{}-K_{k-\tau +1}H\right)}^{T}\Phi^{T}H^{T}K_{k-\tau +2}^{T}{\left(\Phi^{\tau -2}\right)}^{T}-\Phi^{\tau -2}\Gamma Q_{k-\tau +1}\Gamma^{T}H^{T}K_{k-\tau +2}^{T}{\left(\Phi^{\tau -2}\right)}^{T} \end{array}$$
(22)$$\begin{array}{l} P_{k|k-\tau }^{\left(3\right)}=E\left\{({\hat{x}}_{k|k-\tau }-x_{k}){\left[\Phi^{\tau -3}K_{k-\tau +3}(H{\tilde{x}}_{k-\tau +3|k-\tau +2}+v_{k-\tau +3})\right]}^{T}\right\}\\ =E\{\left[\Phi^{\tau -1}({\hat{x}}_{k-\tau +1|k-\tau }-x_{k-\tau +1})-L(w_{k-\tau +1}\cdots w_{k-1})\right]{\left[\Phi^{\tau -3}K_{k-\tau +3}(H{\tilde{x}}_{k-\tau +3|k-\tau +2}+v_{k-\tau +3})\right]}^{T}\}\\ =-\Phi^{\tau -1}P_{k-\tau +1|k-\tau }{\left[\Phi^{2}(I_{n}^{}-K_{k-\tau +1}H)-\Phi K_{k-\tau +2}H\Phi (I_{n}^{}-K_{k-\tau +1}H)\right]}^{T}H^{T}K_{k-\tau +3}^{T}{\left(\Phi^{\tau -3}\right)}^{T}\\ -\Phi^{\tau -2}\Gamma Q_{k-\tau +1}\Gamma^{T}{\left(I_{n}^{}-K_{k-\tau +2}H\right)}^{T}\Phi^{T}H^{T}K_{k-\tau +3}^{T}{\left(\Phi^{\tau -3}\right)}^{T}-\Phi^{\tau -3}\Gamma Q_{k-\tau +2}\Gamma^{T}H^{T}K_{k-\tau +3}^{T}{\left(\Phi^{\tau -3}\right)}^{T}\\ =-\Phi^{\tau -1}P_{k-\tau +1|k-\tau }{\left(A_{3}\right)}^{T}H^{T}K_{k-\tau +3}^{T}{\left(\Phi^{\tau -3}\right)}^{T}-\Phi^{\tau -2}\Gamma Q_{k-\tau +1}\Gamma^{T}{\left(I_{n}^{}-K_{k-\tau +2}H\right)}^{T}\Phi^{T}H^{T}K_{k-\tau +3}^{T}{\left(\Phi^{\tau -3}\right)}^{T}\\ -\Phi^{\tau -3}\Gamma Q_{k-\tau +2}\Gamma^{T}H^{T}K_{k-\tau +3}^{T}{\left(\Phi^{\tau -3}\right)}^{T} \end{array}$$From Equations (20)–(22), the solution rules can be summarized as Equation (15). Similarly, $E\left\{({\hat{x}}_{k|k}-x_{k}){\left({\hat{x}}_{k|k-\tau }-x_{k}\right)}^{T}\right\}$ is defined by:(23)$$E\left\{({\hat{x}}_{k|k}-x_{k}){\left({\hat{x}}_{k|k-\tau }-x_{k}\right)}^{T}\right\}=P_{k|k-\tau }+{\left(P_{k|k-\tau }^{\left(1\right)}\right)}_{}^{T}+{\left(P_{k|k-\tau }^{\left(2\right)}\right)}_{}^{T}+\cdots +{\left(P_{k|k-\tau }^{\left(\tau \right)}\right)}_{}^{T}$$
In summary, the error variance matrix in Equation (18) can be calculated as Equation (12).(3)The proof of $P_{k|k-\tau }$ in Equation (14), from Equation (1):
(24)$$x_{k}=\Phi^{\tau -1}x_{k-\tau +1}+\sum_{i=2}^{\tau }\Phi^{\tau -i}\Gamma w_{k-\tau +i-1},\;\tau \geq 2$$
since $w_{k}$ is white noise and independent of $v_{k}$, the Equation (11) is yielded. Based on $P_{k|k-\tau }=E\{(x_{k}^{}-{\hat{x}}_{k|k-\tau }){\left(x_{k}^{}-{\hat{x}}_{k|k-\tau }\right)}^{T}\}$, Equation (11) and Equation (24), the Equation (14) is yielded. The remaining equations are easily obtained by the Kalman filter and prediction. □


Considering that the test statistic $\gamma_{k}$ obeys the standard Gaussian distribution, under the selected level of significance α, the trigger threshold will be given by using the hypothesis test that satisfies the Gaussian distribution [23,24].

**Corollary** **1.***Under the selected significance level α, set original hypothesis $H_{0}$: ${\hat{x}}_{k|k-\tau }\neq {\hat{x}}_{k|k}$, and antithetic hypothesis $H_{1}$: ${\hat{x}}_{k|k-\tau }={\hat{x}}_{k|k}$. $\gamma_{k,i}$ is the ith element of $\gamma_{k}$, if the observed value satisfies*(25)$$\left|\gamma_{k,i}\right|\leq Z_{\alpha /2},i=1,2,\ldots ,n\;$$*then reject* $H_{0}$ *(non-trigger communication), otherwise accept*  $H_{0}$ *(trigger communication).* $Z_{\alpha /2}$ *is the* α/2 *quantile of the standard Gaussian distribution.*

**Remark****1:** 
*Selecting the original hypothesis $H_{0}:{\hat{x}}_{k|k-\tau }\neq {\hat{x}}_{k|k}$ as the trigger event, the purpose is to protect the trigger event $H_{0}$; thus, priority is given to ensuring the estimation accuracy of the receiving center. On the contrary, if the original hypothesis is the event $H_{1}$: ${\hat{x}}_{k|k-\tau }={\hat{x}}_{k|k}$, this is to protect the event $H_{1}$, so as to give priority to saving the communication of smart nodes. Usually, 1−α is very small, which means that the probability of the following error events ${\bar{A}}_{k,i}$ ($A_{k,i}$ is the trigger event caused by the ith element of the state vector at time k,i=1⋯n) is low:*

(26)
$$P\left\{{\bar{A}}_{k,i}\right\}=P\left\{\left|\gamma_{k,i}\right|\leq Z_{\alpha /2}|{\hat{x}}_{k|k-\tau }\neq {\hat{x}}_{k|k}\right\}=1-\alpha$$



### 2.2. Kalman Filter Algorithm Based on Event-Triggered Mechanism

Since indiscriminate filter calculations and trigger judgments of smart nodes are described in Section 2.1, only the algorithm of the receiving center is introduced in this section. In the receiving center, the Kalman filter algorithm based on the event-triggered mechanism (ET-KF) can be expressed as:(27)$${\hat{x}}_{k}^{}=s_{k}^{}{\hat{x}}_{k|k}^{}+(1-s_{k}^{}){\hat{x}}_{k|k-\tau_{k}^{}}^{},\;k=1,2,\cdots$$
or recursively calculated as:(28)$${\hat{x}}_{k}^{}=s_{k}^{}{\hat{x}}_{k|k}^{}+(1-s_{k}^{})\Phi {\hat{x}}_{k-1}^{},\;k=1,2,\cdots \;$$
where ${\hat{x}}_{k}^{}$ is the final estimate of ET-KF in the receiving center at time k, $k-\tau_{k}^{}$ is the last trigger time before time k, $s_{k}^{}$ is the trigger state of smart sensor nodes at time k; i.e., $s_{k}^{}=1$ means that the transmission event is triggered and the receiving center receives the filter ${\hat{x}}_{k|k}^{}$ of smart sensor nodes, and $s_{k}^{}=0$ means that the transmission event is not triggered and the receiving center will predict, that is
(29)$$s_{k}=\left\{ \begin{array}{l} \begin{array}{cc} 0, & \forall \left|\gamma_{k,i}\right|\leq Z_{\alpha /2}^{},\;i=1,\cdots ,n \end{array}\\ \begin{array}{cc} 1, & \mathrm{otherwise} \end{array} \end{array}\right.\;$$Therefore, the flow chart of ET-KF is shown in Figure 1.

## 3. Performance Analysis of the ET-KF

From Equation (26), the probability $P\{A_{k,i}\}$ of the trigger event is α, and the event $A_{k,i}$ is determined by ${\left|\gamma_{k}\right|}_{i}$, which is the linear combination of measurements $z_{0\sim k}$. When $\mathrm{rank}(z_{k})=r\leq n$, it means that there are n−r events that are completely related to other events. Note that the event-triggered statistic $\gamma_{k}$ obeys the standard Gaussian distribution (for events subject to normal distribution, linear independence is equivalent to mutual independence), that is, $E\{\gamma_{k,i}\gamma_{k,j}^{}\}=0,\;i\neq j$ and $E\{\gamma_{k,i}\gamma_{k,i}^{}\}=1$, then the probability $P\{A_{k,i}A_{k,j}\}=0,\;i\neq j$. Selecting r linearly unrelated events $A_{k,1},A_{k,2}\ldots A_{k,r}$, according to the trigger mechanism in (27)–(29), the event-triggered probability can be calculated by:(30)$$p=P\{A_{k,1}\cup A_{k,2}\cup \ldots \cup A_{k,r}\}=1-P\{{\bar{A}}_{k,1}{\bar{A}}_{k,2}\ldots {\bar{A}}_{k,r}\}\;=1-P\{{\bar{A}}_{k,1}\}P\{{\bar{A}}_{k,2}\}\ldots P\{{\bar{A}}_{k,r}\}=1-{\left(1-\alpha \right)}^{r}$$

Therefore, $P\{s_{k}=1\}=p$, $P\{s_{k}=0\}=1-p$. The estimated error variance $P_{k|k}$ of the ET-KF in the receiving center is given by the following theorem.

**Theorem** **2.***The estimation error variance*
 $P_{k}^{}$ *of the ET-KF in the receiving center can be calculated as:**(1) If the system in Equations (1) and (2) has a steady state, that is,* $P_{k}$ *has a steady state value* $\bar{P}$*, then* $P_{k}$ *can be calculated by the following Riccati equation:*(31)$$\sum =p\bar{P}+(1-p)\Phi \sum \Phi_{}^{T}+(1-p)\Phi \Gamma Q_{k-1}\Gamma_{}^{T}+(1-p)\Gamma Q_{k-1}\Gamma_{}^{T}\Phi_{}^{T}+(1-p)\Gamma Q_{k-1}\Gamma_{}^{T}$$*where*(32)$$\underset{k\to \infty }{\lim}P_{k}=\sum$$*(2) If the system is a time-varying system,*
 $P_{k}$ *can be approximately calculated as:*
(33)$$P_{k}\approx pP_{k|k}+(1-p)pP_{k|k-1}$$

**Proof.** If the system has a steady state estimation and $P_{k|k}$ has a stationary value $\bar{P}$, then based on Equation (28), $P_{k}$ can be calculated as:
(34)$$\begin{array}{l} P_{k}=E\{{\tilde{x}}_{k}{\tilde{x}}_{k}^{T}\}=E\{(x_{k}-{\hat{x}}_{k}){\left(\cdot \right)}^{T}\}=E\left\{\left(s_{k}(x_{k}-{\hat{x}}_{k|k})+(1-s_{k})(x_{k}-\Phi {\hat{x}}_{k-1})\right){\left(\cdot \right)}^{T}\right\}\\ =E\left\{\left(s_{k}(x_{k}-{\hat{x}}_{k|k})+(1-s_{k})(\Phi x_{k-1}+\Gamma w_{k-1}^{}-\Phi {\hat{x}}_{k-1})\right){\left(\cdot \right)}^{T}\right\}\\ =E\left\{\left(s_{k}{\tilde{x}}_{k|k}+(1-s_{k})(\Phi {\tilde{x}}_{k-1}+\Gamma w_{k-1}^{})\right){\left(\cdot \right)}^{T}\right\} \end{array}$$
where ‘(⋅)’ means the same as the previous formula. Since $s_{k}$ obeys the Bernoulli distribution,
(35)$$E\left\{s_{k}(1-s_{k}){\tilde{x}}_{k|k}{\left(\Phi {\tilde{x}}_{k-1}^{}\right)}_{}^{T}\right\}=0$$Considering that $w_{k}^{}$ is white noise, from Equation (35) and $E\{s_{k}^{2}=1\}=p$, $E\{{\left(1-s_{k}^{}\right)}_{}^{2}\}=1-p$, there are:(36)$$\begin{array}{l} P_{k}=E\{s_{k}^{2}{\tilde{x}}_{k|k}^{}{\tilde{x}}_{k|k}^{T}\}+E\{{\left(1-s_{k}\right)}_{}^{2}\Phi {\tilde{x}}_{k-1}^{}{\tilde{x}}_{k-1}^{T}\Phi_{}^{T}\}+E\{{\left(1-s_{k}\right)}_{}^{2}\Phi \Gamma w_{k-1}^{}w_{k-1}^{T}\Gamma_{}^{T}\}\\ +E\{{\left(1-s_{k}\right)}_{}^{2}\Gamma w_{k-1}^{}w_{k-1}^{T}\Gamma_{}^{T}\Phi_{}^{T}\}+E\{{\left(1-s_{k}\right)}_{}^{2}\Gamma w_{k-1}^{}w_{k-1}^{T}\Gamma_{}^{T}\}\\ =pP_{k|k}+(1-p)\Phi P_{k-1}\Phi_{}^{T}+(1-p)\Phi \Gamma Q_{k-1}\Gamma_{}^{T}+(1-p)\Gamma Q_{k-1}\Gamma_{}^{T}\Phi_{}^{T}+(1-p)\Gamma Q_{k-1}\Gamma_{}^{T} \end{array}$$
the limit value exists in Riccati Equation (36):(37)$$\underset{k\to \infty }{\lim}P_{k}=\sum$$
where ∑ is the solution of the following steady-state Riccati Equation:(38)$$\sum =p\bar{P}+(1-p)\Phi \sum \Phi_{}^{T}+(1-p)\Phi \Gamma Q_{k-1}\Gamma_{}^{T}+(1-p)\Gamma Q_{k-1}\Gamma_{}^{T}\Phi_{}^{T}+(1-p)\Gamma Q_{k-1}\Gamma_{}^{T}$$If the system is a time-varying system, from Equation (28):(39)$${\hat{x}}_{k}=s_{k}{\hat{x}}_{k|k}+(1-s_{k})\Phi [s_{k-1}{\hat{x}}_{k-1|k-1}+(1-s_{k-1})\Phi {\hat{x}}^{}{}_{k-2}]\;=s_{k}{\hat{x}}_{k|k}+(1-s_{k})s_{k-1}{\hat{x}}_{k|k-1}+(1-s_{k})(1-s_{k-1})\Phi_{}^{2}{\hat{x}}^{}{}_{k-2}$$
the error at time k is:(40)$${\tilde{x}}_{k}=x_{k}-{\hat{x}}_{k}=s_{k}(x_{k}-{\hat{x}}_{k|k})+(1-s_{k})s_{k-1}(x_{k}-{\hat{x}}_{k|k-1})+(1-s_{k})(1-s_{k-1})(x_{k}-\Phi_{}^{2}{\hat{x}}^{}{}_{k-2})\;=s_{k}{\tilde{x}}_{k|k}+(1-s_{k})s_{k-1}{\tilde{x}}_{k|k-1}+(1-s_{k})(1-s_{k-1})(x_{k}-\Phi_{}^{2}{\hat{x}}^{}{}_{k-2})$$
since
(41)$$E\left\{s_{k}(1-s_{k})s_{k-1}{\tilde{x}}_{k|k}{\left({\tilde{x}}_{k|k-1}\right)}_{}^{T}\right\}=0\;E\left\{{\left(1-s_{k}\right)}_{}^{2}s_{k-1}(1-s_{k-1}){\tilde{x}}_{k|k-1}{\left(x_{k}-\Phi_{}^{2}{\hat{x}}^{}{}_{k-2}\right)}_{}^{T}\right\}=0\;E\left\{s_{k}(1-s_{k})(1-s_{k-1}){\tilde{x}}_{k|k}{\tilde{x}}_{k|k}{\left(x_{k}-\Phi_{}^{2}{\hat{x}}^{}{}_{k-2}\right)}_{}^{T}\right\}=0$$
 $P_{k}$ can written as:(42)$$P_{k}=E\{{\tilde{x}}_{k}{\tilde{x}}_{k}^{T}\}=E\left\{s_{k}^{2}{\tilde{x}}_{k|k}{\tilde{x}}_{k|k}^{T}\right\}+E\{{\left(1-s_{k}\right)}_{}^{2}s_{k-1}^{2}{\tilde{x}}_{k|k-1}^{}{\tilde{x}}_{k|k-1}^{T}\}\;+E\left\{{\left(1-s_{k}\right)}_{}^{2}{\left(1-s_{k-1}\right)}_{}^{2}(x_{k}-\Phi_{}^{2}{\hat{x}}^{}{}_{k-2}){\left(\cdot \right)}^{T}\right\}\;=pP_{k|k}+(1-p)pP_{k|k-1}+{\left(1-p\right)}^{2}E\left\{(x_{k}-\Phi_{}^{2}{\hat{x}}^{}{}_{k-2}){\left(\cdot \right)}^{T}\right\}$$
since (1−p) is usually very small ($p=1-{\left(1-\alpha \right)}^{r}$), the third term of Equation (42) is omitted, and there are:(43)$$P_{k}\approx pP_{k|k}+(1-p)pP_{k|k-1}$$
□

**Remark** **2.**
*The estimation error variance $P_{k}$ in the receiving center under the event-triggered threshold $Z_{\alpha /2}$ is given in Theorem 2. That is, under the condition that the system has a steady-state filter, $P_{k}$ can be computed by Equation (31) and Equation (32);on the contrary, the approximate calculation is (33). In addition, for a system with steady-state estimation, we can set the threshold reasonably according to the desired theoretical accuracy $P_{k}$. It can be seen from Equations (6) to (13) and (31) to (33) that the final estimation accuracy $P_{k}$ is not only related to the trigger rate, but also directly proportional to the system noise statistics $Q_{k}$ and measurement noise statistics $R_{k}$.*


## 4. Simulation Examples

Scenario 1.(Event-triggered steady-state Kalman filter.) Consider the following planar tracking system:

(44)$$x_{k}=\left[\begin{array}{llll} 1 & T & 0 & 0\\ 0 & 1 & 0 & 0\\ 0 & 0 & 1 & T\\ 0 & 0 & 0 & 1 \end{array}\right]x_{k}+\left[\begin{array}{ll} 0.5T^{2} & 0\\ T & 0\\ 0 & 0.5T^{2}\\ 0 & T \end{array}\right]w_{k},\;k=1,\cdots ,l$$
where $x_{k}={\left[x_{k}{\dot{x}}_{k}y_{k}{\dot{y}}_{k}\right]}^{T}$ is the state of target movement, $x_{k}$, $y_{k}$, ${\dot{x}}_{k}$ and ${\dot{y}}_{k}$ are the positions and velocities of the target in x-axis and y-axis, respectively. The measurement equation of the system is:(45)$$z_{k}=\left[\begin{array}{llll} 1 & 0 & 0 & 0\\ 0 & 0 & 1 & 0 \end{array}\right]x_{k}+v_{k},\;k=1,\cdots ,l$$

The estimation performance is mean square error (MSE) [25,26]:(46)$${\mathrm{MSE}}_{k}^{}=\frac{1}{k}\sum_{t=0}^{k}\frac{1}{N}\sum_{j=1}^{N}{\left(x_{t}^{j}-{\hat{x}}_{t}^{j}\right)}^{2},\;k=1,\cdots ,l$$
where $x_{t}^{j}$ and ${\hat{x}}_{t}^{j}$ are the true states and their estimations of the jth Monte Carlo experiment at time t.

In the simulation, the working period is T=0.5 s and the length of work is l=200 steps, the variance of processing noise is $Q=diag(0.2\;m^{2}{/s}^{4},0.2\;m^{2}{/s}^{4})$, and the variance of measurement noise is $R=diag(0.4\;m^{2}{/s}^{4},0.4\;m^{2}{/s}^{4})$, the initial state is $x_{0}=$${\left[10\;m,\;1\;m/s,\;10\;m,\;1\;m/s\right]}^{T}$. A total of 200 Monte Carlo experiments are carried out. Because the measurement Equation (45) is a two-dimensional system and only the positions on the x-axis and y-axis are measured, according to Equation (26), the theoretical trigger probability is $p=1-{\left(1-\alpha \right)}^{2}$. Here, α=0.98, 0.8, 0.6, 0.4 are chosen for comparison; that is, the thresholds are 0.05, 0.25, 0.52, and 0.84; and the theoretical trigger rates are 0.9996, 0.96, 0.84, and 0.64. For the threshold $Z_{\alpha /2}$, the theoretical trigger probability and the actual trigger frequency are shown in Table 1. The theoretical trigger probability is consistent with the result of the actual trigger frequency, which verifies the correctness of Theorem 1 and Corollary 1. It shows that, when the proposed trigger mechanism is used, the actual trigger frequency can be predicted, and communication resources can be adjusted in advance to reduce the possibility of network congestion.

The error covariance $P_{k}$ calculated by Equation (31) and ${\mathrm{MSE}}_{k}^{}$ obtained by Equation (46) are shown in Table 2. It can be seen that the theoretical accuracy (diagonal elements of error covariance matrix $P_{k}$) is close to ${\mathrm{MSE}}_{k}^{}$ obtained in actual work, which verifies the correctness of theorem 2 under the steady-state situation. When adopting the proposed trigger mechanism, the proposed ET-KF accuracy can be pre-calculated, and then the most suitable trigger frequency can be obtained by a reasonable trigger threshold.

$P_{k,i}(i=1,\cdots ,n)$ and ${\mathrm{MSE}}_{200,i}$ is the i,i diagonal element in the error covariance matrix $P_{k}$ of the ET-KF, and ${\mathrm{MSE}}_{200,i}$ is the i element of ${\mathrm{MSE}}_{200}^{}$ for the ET-KF.Scenario 2. (Event-triggered time-varying Kalman filter.) Consider the planar tracking system in Equation (44) and Equation (45), where $x_{0}=\left[10\;m\;1\;m/s\;10\;m\;1\;m/s\right]$ is the initial state, $Q_{k}=\mathrm{diag}(0.2+0.2\sin(2\pi k/100)m^{2}{/s}^{4},\;\;0.2+0.2\sin(2\pi k/100)m^{2}{/s}^{4})$ is processing noise variance, $R_{k}=\mathrm{diag}(0.3+0.5\sin(2\pi k/100)m^{2}{/s}^{4}\;,0.3+0.5\sin(2\pi k/100)m^{2}{/s}^{4})$ is measurement noise variance. The performance of the system is measured by the accumulated mean square error (AMSE) [27,28,29]:

(47)$${\mathrm{AMSE}}_{k}^{}=\sum_{t=1}^{k}\frac{1}{N}\sum_{j=1}^{N}{\left(x_{t}^{j}-{\hat{x}}_{t}^{j}\right)}_{}^{T}(x_{t}^{j}-{\hat{x}}_{t}^{j})$$
where $x_{t}^{j}$ and ${\hat{x}}_{t}^{j}$ are the true states and their estimations of the jth Monte Carlo experiment at time t. Corresponding to AMSE, the theoretical estimation accuracy is measured by $\Sigma \mathrm{tr}P_{k}^{}$:(48)$$\Sigma \mathrm{tr}P_{k}^{}=\sum_{t=1}^{k}\frac{1}{N}\sum_{j=1}^{N}\mathrm{tr}(P_{t}^{j})$$
where $\mathrm{tr}(P_{t}^{j})$ is the trace of the estimated error covariance matrix at time t of the jth Monte Carlo experiment.

In Table 3, it is shown that for time-varying systems, the theoretical trigger frequency is close to the actual trigger frequency, which verifies the correctness of Theorem 1 and Corollary 1.

In Figure 2, with a different α, the comparison curves of AMSE of ET-KF are shown. It can be seen that the smaller α is, the greater the threshold $Z_{\alpha /2}$ is, the lower the trigger rate is, and the worse the accuracy AMSE is. Comparison curves between the theoretical approximate estimation accuracy $\Sigma \mathrm{tr}P_{k}^{}$ (the sum of trace of $P_{k}^{}$) calculated by (33) and the AMSE obtained by the experiment with α=0.60 are shown in Figure 3. It can be seen that the theoretical estimation accuracy is close to the AMSE obtained by the experiment, which verifies the correctness of Theorem 2 under the time-varying situation. The AMSEs in Figure 2 and Figure 3 rise in a straight line, which means that the corresponding MSEs are stable.

In Table 4 and Figure 4, ET-KF is compared with the event trigger of innovation standardization (IS-KF) [14], a dynamic event trigger constructed by the linear change in the rate of change (LC-KF) [22]. From Table 4, under the same trigger rate (80%), the ${\mathrm{AMSE}}_{200}$ (AMSE at step 200) of the proposed trigger mechanism is lower than the other two trigger mechanisms. Figure 4 shows the comparison curve of AMSE at different steps, as the number of steps increases, the AMSE value gap based on different trigger mechanisms becomes larger. Among them, the AMSE value of the ETKF mechanism is the smallest and the estimation accuracy is the highest, which verifies the correctness of the conclusion.

## 5. Conclusions

In estimation, for linear systems, an event-triggered Kalman filter (ET-KF) is proposed in this paper. The main work is as follows:The event-triggered statistic is constructed, which proves that the statistic obeys the standard Gaussian distribution, according to the event-triggered statistic and hypothesis test of the Gaussian distribution, the significance of the event-triggered threshold is given, and then an event-triggered estimation mechanism is designed.Based on the event-triggered threshold and mechanism proposed in this paper, the theoretical trigger frequency under different thresholds and the estimation accuracy of event-triggered systems are analyzed. The proposed ET-KF can accurately set the event trigger frequency in advance. For linear systems with steady-state estimation, the estimation accuracy can be obtained by the Riccati equation accurately, and the trigger threshold can be set according to the accuracy. For linear time-varying systems without steady-state estimation, the approximate estimation accuracy can be obtained.The proposed trigger mechanism has higher estimation accuracy at the same trigger rate, and the trigger setting is reasonable.

## Figures and Tables

**Figure 1 sensors-23-02202-f001:**
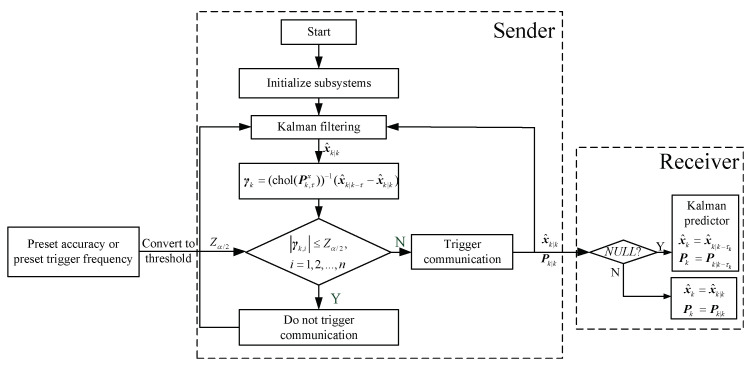
The flow chart of ET-KF.

**Figure 2 sensors-23-02202-f002:**
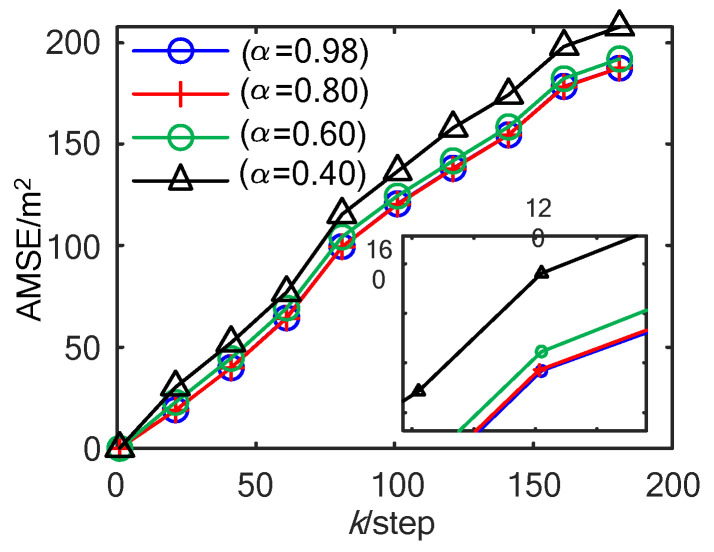
AMSE of the ET-KF algorithm with different α.

**Figure 3 sensors-23-02202-f003:**
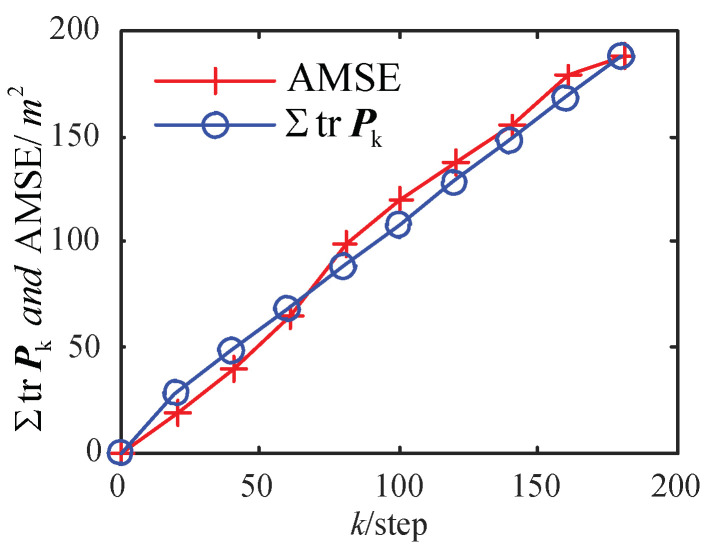
AMSE of ET-KF algorithm and $\Sigma \mathrm{tr}P_{k}^{}$ with α=0.60.

**Figure 4 sensors-23-02202-f004:**
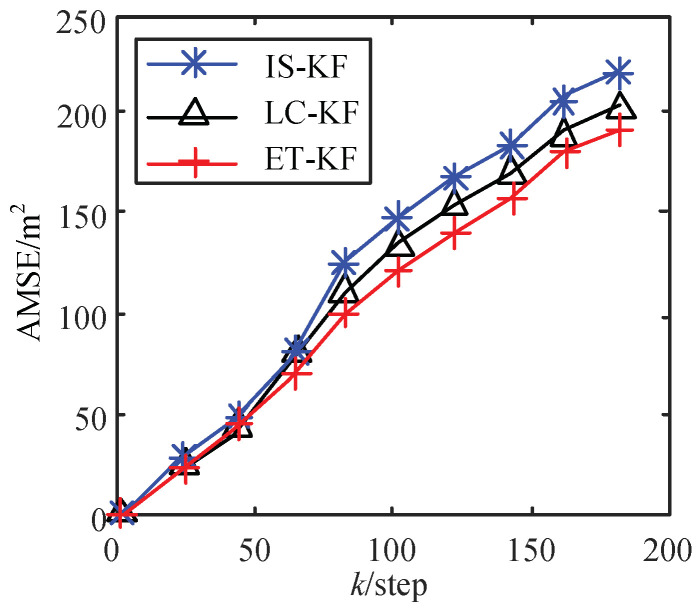
AMSE of different trigger threshold algorithms.

**Table 1 sensors-23-02202-t001:** Comparison of trigger frequency in the steady-state system for different α.

α	0.98	0.8	0.6	0.4
Threshold $Z_{\alpha /2}$	0.05	0.25	0.52	0.84
Theoretical trigger probability	0.9996	0.96	0.84	0.64
Actual trigger frequency	99.5%	96%	83.5%	63%

**Table 2 sensors-23-02202-t002:** The error covariance of the steady-state ET-KF algorithm and ${\mathrm{MSE}}_{i}$.

	$P_{k,1}$	${\mathrm{MSE}}_{200,1}$	$P_{k,2}$	${\mathrm{MSE}}_{200,2}$	$P_{k,3}$	${\mathrm{MSE}}_{200,3}$	$P_{k,4}$	${\mathrm{MSE}}_{200,4}$
α=0.98	0.1795	0.1790	0.1450	0.1544	0.1795	0.1816	0.1450	0.1345
α=0.80	0.1848	0.1792	0.1450	0.1548	0.1848	0.1807	0.1450	0.1342
α=0.60	0.2084	0.1835	0.1450	0.1594	0.2084	0.1849	0.1450	0.1365
α=0.40	0.2614	0.1988	0.1450	0.1662	0.2614	0.1850	0.1450	0.1362

**Table 3 sensors-23-02202-t003:** Comparison of trigger frequency in the time-varying system under different α values.

α	0.98	0.8	0.6	0.4
Threshold $Z_{\alpha /2}$	0.05	0.25	0.52	0.84
Theoretical trigger probability	0.9996	0.96	0.84	0.64
Actual trigger frequency	99.5%	95.5%	82%	65%

**Table 4 sensors-23-02202-t004:** The performance comparison of different trigger threshold algorithms.

Different Trigger Threshold Algorithms	${\mathrm{AMSE}}_{200}$	Trigger Frequency
ET-KF	198	80%
IS-KF	225	80%
LC-KF	210	80%

## Data Availability

Not applicable.

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
