# Peer review of "Event-Triggered Kalman Filter and Its Performance Analysis"

_sensors, 2023, doi:10.3390/s23042202_

Round 1
Reviewer 1 Report
An event‐triggered Kalman filter (ET‐KF) is proposed in this paper by the authors. The authors claimed that, "The proposed trigger mechanism has higher estimation accuracy at the same trigger rate, and the trigger setting is reasonable". The authors are advised to justify the same by comparing with more existing algorithms and need better evidence to claim the same.
Reviewer 2 Report
in this paper, the author(s) proposed theCombining of the threshold and the event‐triggered mechanism, an event‐triggered Kalman mechanism to approximate estimation accuracy can be calculated. The advantage of the proposed mechanism is whether it 1is a steady system or a time‐varying system, the proposed algorithm can reasonably set the threshold according to the required accuracy in advance. Compared with other event‐triggered mechanisms, the proposed event‐triggered estimator not only effectively reduces the communication cost, but also has higher accuracy.
The introduction section is well written, methodlogy is well explained.
I have few suggestions which will improve the overall quality of the paper.
1) add detailed related work section after introduction, their merits and demerits, and how the proposed system is overcoming the problems in existing system(s).
2) add system diagram or atleast a flow chart to show the inputs and outputs.
3) results need more explaination and detailed discussion.
Reviewer 3 Report
This paper presents a novel event triggered Kalman filter for linear systems. The filter uses a Gaussian distribution hypothesis test to determine the event-triggered threshold, which allows for accurate control of trigger frequency. By combining the threshold with the event-triggered mechanism, the filter provides high accuracy in estimation, regardless of whether the system is steady or time-varying. The proposed algorithm can be tailored to meet desired accuracy requirements. The paper claims that the algorithm outperforms other event-triggered mechanisms, providing both lower communication cost and improved accuracy. The results of the simulation examples support the validity and effectiveness of the proposed algorithm. However, it needs to address the following issues.
i) The proposed method uses the hypothesis test in the Kalman filter for triggering an event. However, the related work on hypothesis test is missing. The author should discuss the related work on using hypothesis test (see the following articles) in the Kalman filter to strengthen the literature survey.
* A hypothesis test-constrained robust Kalman filter for INS/GNSS integration with abnormal measurement, IEEE Transactions on Vehicular Technology, 2023.
* A hypothesis test-constrained robust Kalman filter for INS/GNSS integration with abnormal measurement, IEEE Transactions on Vehicular Technology, 2022.
* Double-channel sequential probability ratio test for failure detection in multi-sensor integrated systems, IEEE Transactions on Instrumentation and Measurement, Vol. 70, 2021.
ii) Page 5: the symbol "chol" should keep the text formatting consistent, and the decomposition can be omitted in this case.
iii) Page 4, Algorithm 1: the symbol ‘’, appears without explanation in previous context, though it seems to be a sequential index. The author should identify its meaning.
iv) Page 5, equation 16: the superscript ‘*’ appears without any explanation, the authors should identify its meaning.
v) Page 3: the measurement noise covariance R in equation 7 should have the subscript k, and the authors should identify the meaning of symbols in Theorem 1.
vi) Fig.1: the subgraph should mark the position or the coordinate.
vii) From the simulation result it can be obvious that the accumulated mean square error is increasing, which may degrade the estimate performance. It is suggested that the authors discuss some methods to eliminate the influence, which can be referred to [1] and [2].
* Robust adaptive filtering method for SINS/SAR integrated navigation system, Aerospace Science and Technology, Vol. 15, 2011.
* Extended Kalman filter for online soft tissue characterization based on Hunt-Crossley contact model, Computers in Biology and Medicine, Vol. 137, 2021.
Reviewer 4 Report
1. The article title should be improved by focusing on the aim of the study and also using scientific terminology.
2. A comparative analysis of previous studies must present limitations. As it is difficult to justify the novelty of the study without discussing the limitations of previous studies.
3. The methodology of the study must be included in pictorial representation for better readability. Mention, how the proposed methodology is novel when compared with previous methodologies.
4. The abstract of the article is starting with “For linear systems”. It clearly concludes that the authors lack knowledge about scientific writing in a research article. In the abstract, it is mentioned in which field, the event trigger algorithm is proposed for analysis. Along with highlighting the findings of the simulation in the abstract for concluding the novelty of the study. Don’t write the statement like this “Compared with other event‐triggered mechanisms, the proposed event‐triggered estimator not only effectively reduces the communication cost but also has higher accuracy” without any strong finding to it.
5. The study has missed high quality articles that focused on event trigger and its implementation. The following are the articles:
https://doi.org/10.1109/TCST.2016.2623776; https://doi.org/10.1109/TAC.2015.2492159
https://doi.org/10.1109/TCYB.2016.2523878
https://doi.org/10.1016/j.ins.2022.04.033
Round 2
Reviewer 1 Report
The paper has been revised well as per my previous comments and the improved version is satisfactory. Paper shall be accepted in present form.
Author Response
Thank you very much for your comments.
Reviewer 4 Report
most of the comments addressed by the authors, so no further comments.
Author Response
Thank you very much for your comments.